# CD169^+^ Macrophages in Primary Breast Tumors Associate with Tertiary Lymphoid Structures, T_regs_ and a Worse Prognosis for Patients with Advanced Breast Cancer

**DOI:** 10.3390/cancers15041262

**Published:** 2023-02-16

**Authors:** Oscar Briem, Eva Källberg, Siker Kimbung, Srinivas Veerla, Jenny Stenström, Thomas Hatschek, Catharina Hagerling, Ingrid Hedenfalk, Karin Leandersson

**Affiliations:** 1Cancer Immunology, Department of Translational Medicine, Lund University, 214 28 Malmö, Sweden; 2Division of Oncology, Department of Clinical Sciences, Lund University, 223 81 Lund, Sweden; 3Division of Clinical Genetics, Department of Laboratory Medicine Lund, Lund University, 221 84 Lund, Sweden; 4Department of Oncology and Pathology, Karolinska Institutet, 171 77 Solna, Sweden

**Keywords:** breast cancer, lymph node, macrophage, CD169, T_reg_, B_reg_, TLS

## Abstract

**Simple Summary:**

We here show that CD169^+^ TAMs in primary breast tumors are associated with tertiary lymphoid-like structures (TLLSs), T_reg_ and B_reg_ signatures, and a worse prognosis for the patient. In contrast, CD169^+^ TAMs and TLLSs present in lymph node metastases were associated with better prognosis. We propose that the negative prognostic value related to CD169^+^ TAMs and TLLSs in primary breast tumors is a unique consequence of an immunosuppressive tumor environment in advanced breast cancers. This knowledge is important for understanding the immune landscape in breast cancer and for future targeted therapies.

**Abstract:**

The presence of CD169^+^ macrophages in the draining lymph nodes of cancer patients is, for unknown reasons, associated with a beneficial prognosis. We here investigated the prognostic impact of tumor-infiltrating CD169^+^ macrophages in primary tumors (PTs) and their spatial relation to tumor-infiltrating B and T cells. Using two breast cancer patient cohorts, we show that CD169^+^ macrophages were spatially associated with the presence of B and T cell tertiary lymphoid-like structures (TLLSs) in both PTs and lymph node metastases (LNMs). While co-infiltration of CD169^+^/TLLS in PTs correlated with a worse prognosis, the opposite was found when present in LNMs. RNA sequencing of breast tumors further confirmed that *SIGLEC1* (CD169) expression was associated with mature tertiary lymphoid structure (TLS), and T_reg_ and B_reg_ signatures. We propose that the negative prognostic value related to CD169^+^ macrophages in PTs is a consequence of an immunosuppressive tumor environment rich in TLSs, T_regs_ and B_regs_.

## 1. Introduction

Breast cancer is a high-impact disease in our society. With a high mortality rate, due to metastasis, breast cancer is the fifth deadliest cancer type worldwide and even passed lung cancer in incidence rate in 2020 [1]. The need for novel therapies and improvement in current treatment regimens is urgent.

In general, breast cancers are divided into various subtypes depending on hormone receptor expression status (estrogen receptor, ER, and progesterone receptor, PR) and human epidermal growth factor receptor 2 (HER2) status. Expression of these receptors has a large impact on choice of current treatment protocols and on breast cancer prognosis. While receptor positive breast cancers are more common (ER^+/−^PR^+/−^HER2^+/−^), triple negative breast cancers (ER^−^PR^−^HER2^−^; TNBC) are less common and have the worst prognosis with few treatment options [2,3].

In breast cancer, the response rate to immune checkpoint blockade is still relatively low [4,5]. While immune checkpoint inhibitors focus on promoting cytotoxic T-cell activation, other immune cell populations infiltrating the tumor microenvironment (TME) are being further investigated in order to increase our understanding and the efficacy of current treatments [6,7]. Among the most important immune populations in the TME are macrophages and the myeloid immune cell compartment [8].

Macrophages are innate myeloid immune cells with a wide plasticity. They are broadly divided into either tissue-resident macrophages or recruited monocyte-derived macrophages [9]. Apart from this division, macrophage subsets are further characterized by their polarization state. There are two extreme macrophage polarization states, often being referred to as M1- and M2-like subsets, with a plethora of subpopulations ranging in between them, depending on localization, microenvironment and the type of disease in which they are active [10,11].

Lately, a tissue-resident macrophage subpopulation with expression of the surface marker CD169^+^ has been attracting attention, due to its highly prognostic impact in cancer and autoimmune disease [12]. CD169^+^ is expressed and upregulated predominantly on macrophages found in organs such as lungs, bone marrow and secondary lymphoid organs (SLOs) [13]. In the SLOs, the CD169^+^ macrophages are either subcapsular sinus (SCS) CD169^+^ macrophages or medullary CD169^+^ macrophages, with slightly different origin and function [14,15]. Their main function there is associated with lymphoid cell activation and regulation [16,17]. While the SCS CD169^+^ macrophages capture opsonized antigens or lymph-born antigens, allowing antigen encounters with underlying B-cell follicles and thus inducing a germinal center B-cell response, [18] the medullary sinus CD169^+^ macrophages are efficient at phagocytosis, pathogen clearance, sensing lipids and inducing tissue destruction [18,19]. In a tumor context, CD169^+^ macrophages can originate from activated monocytes [20] that infiltrate tumors, hence becoming tumor-associated macrophages (TAMs) [21].

In 2012, it was reported that CD169^+^ macrophages located in the paracortical region of lymph nodes were able to catch tumor antigens and use cross-presentation to activate CD8 T-cells [16]. It was also shown that SCS CD169^+^ macrophages could recognize sialic acid decorated apoptotic bodies from tumor cells, facilitating B cell anti-tumor immunity [22]. These initial findings were followed by several cohort studies presenting evidence that high presence of CD169^+^ macrophages in lymph nodes of cancer patients was associated with a beneficial prognosis [23,24,25,26]. We recently confirmed this phenomenon in breast cancer and showed that the presence of CD169^+^ macrophages in breast cancer lymph node metastasis (LNM) was associated with a better prognosis, while surprisingly the presence of CD169^+^ tumor-associated macrophages (CD169^+^ TAMs) in the primary tumor (PT) was not [27].

The functional localization of CD169^+^ macrophages surrounding lymphocyte follicles in SLOs led us to speculate whether infiltrating CD169^+^ macrophages in PTs (CD169^+^ TAMs) would localize with tumor-infiltrating lymphocytes (TILs). We specifically investigated whether the CD169^+^ TAMs in PTs would localize to tertiary lymphoid structures (TLS) or tertiary lymphoid-like structures (TLLSs), similar to the spatial positions they have in secondary lymphoid follicle structures, and the prognostic effect thereof. We here provide evidence that CD169^+^ TAMs associate with TLLS, T_reg_ and B_reg_ signatures in breast cancers, leading to an adverse clinical outcome when present in PTs, while the opposite effect was observed in LNMs.

## 2. Materials and Methods

### 2.1. Ethical Declarations

Written informed consent was received from the patients included in the clinical trials presented in this study, and ethical approvals for the clinical trials were obtained from the regional ethics committees in Sweden: Stockholm (Dnr KI 02-206 and KI 02-205) and Lund (Dnr 2009/658) [28,29].

### 2.2. Patient Cohorts and Study Design

Two patient cohorts were used in this study; the first cohort was a retrospective cohort study based on primary tumors and lymph node metastases from patients with locally advanced and metastatic breast cancer from the randomized phase III TEX trial performed between 2002–2007 [28]. Detailed information about the clinical trial is found at clinicaltrials.gov with identification number NCT01433614. Briefly, the clinical trial comprised 304 women with advanced or inoperable metastatic breast cancer. Participants received two types of combination chemotherapy as the first line of treatment: Epirubicin and Paclitaxel alone or combined with Capecitabine. Among several criteria that have been described in detail previously [28], the enrolled participants had to have a life expectancy of at least 3 months, no brain metastases and they were not permitted to join the study if they had performed previous chemotherapy treatment cycles. From the 304 participants, formalin-fixed, paraffin-embedded blocks from primary tumors and synchronous lymph node metastases were collected wherever possible for tissue microarray (TMA) construction, as described previously [29], enabling further analysis of the tissue with immunohistochemistry (IHC). A simplified study design of the cohort is illustrated in Figure 1A.

The second cohort was a broad prospective, population-based cohort used in order to validate our findings from the smaller cohort using RNA sequencing data. This cohort was compromised of 8164 patients enrolled in the Sweden Cancerome Analysis Network—Breast (SCAN-B) initiative [30], and was approved by the regional ethical review board in Lund, Sweden. Detailed information from the cohort is found at ClinicalTrials.gov with identification number NCT02306096. Fresh biopsy samples were taken from each patient during the primary surgery by pathologists performing their routine clinical diagnostics. All analyses were performed in accordance with patient consent and ethical regulations, and the biopsies were used to gather RNA sequencing data.

### 2.3. Immunohistochemistry and Scoring

IHC was performed on the TMA cohort where all primary tumor and lymph node metastases were scored for the different immune cell surface markers, CD169, CD20 and CD3, using the protocol previously described [29,31]. FoxP3 had been annotated previously [31]. In brief, TMA blocks were sectioned to a thickness of 4 mm prior to mounting. The sections contained cores with diameters of 800 μm and were pre-treated with the PT-link system before staining with an Autostainer Plus (DAKO, Santa Clara, CA, USA) at pH6 with an overnight staining protocol. The following antibodies and dilutions were used for staining: anti-CD169^+^ macrophages (1:100, Invitrogen, Clone SP216, Waltham, MA, USA), anti-CD20^+^ B-cells (1:100, Abcam, Clone L-26, Cambridge, UK), anti-CD3^+^ T-cells (1:100, Abcam, Clone 11084, Cambridge, UK) and developed with a triple staining IHC kit from Abcam. A previous staining performed by authors (J.S. and C.H.) used mouse monoclonal anti-FoxP3 (ab20034, clone 236A/E7, Abcam, 1:400, Cambridge, UK) to annotate T_regs_ as previously published [31].

CD3^+^ T-cells, CD20^+^ B-cells and CD169^+^ macrophages were annotated individually by authors O.B., E.K. and K.L. The following scores were used: for CD169, CD169^+^ expression present = 1, and CD169^+^ expression absent = 0; for CD20, CD20^+^ clusters in spatial contact with CD3^+^ T-cells present = 1, and CD20^+^ absent or present as dispersed single cells (not in clusters) or without spatial contact with CD3^+^ T-cells = 0. Since we did not include a follicular dendritic cell marker, the CD20^+^/CD3^+^ B/T cell clusters will be referred to as TLLS, and not TLS. The purpose of this scoring was to classify immune cell infiltration into three different categories: (1) CD169^+^ macrophages positive tumors/metastases; (2) TLLS positive tumors/metastases (CD20^+^/CD3^+^); (3) tumors/metastases with presence of CD169^+^ and TLLS (CD169^+^/CD20^+^/CD3^+^), as represented in Figure 1B.

T_reg_ (FoxP3^+^) annotations were published previously and performed by authors J.S. and C.H. [31]. The T_regs_ scoring strategy ranged from 0–3 and furthermore also categorized absence–presence (0–1). In the present study, the T_regs_ (FoxP3^+^) (0–1) score was used, solely exploring correlation between presence or absence of T_regs_ and its effect within the three different immune cell infiltration categories (CD169^+^, TLLS, CD169^+^/TLLS).

### 2.4. Statistical Analyses

Statistical analyses were performed with IBM SPSS statistics (version 27), with all statistical tests being two-sided with *p* ≤ 0.05 considered as significant results. In the TMA cohort, age at diagnosis ranged from 27 to 71 years old with an overall median age at diagnosis of 51 years. A total of 21% of included patients were alive at the time of data collection (July 2013) and the median follow-up time for patients alive was 10.5 years. Age, tumor size, lymph node status, metastatic stage, PT receptor status, lymph node receptor status and adjuvant therapy given are presented in Table 1.

Correlations between clinicopathological factors and immune cell infiltration in PTs and LNMs were assessed using cross tabulation tables. Odds ratios with a 95% confidence interval were correlated to 5-year recurrence-free interval (RFI), 5-year breast-cancer-specific survival (BCSS), tumor sizes above 20 mm, expression of the receptors (ER, PR, HER2), high Ki67 levels (>15%) and presence of T_regs_, TLLSs or CD169^+^ macrophages. All clinicopathological factors were set as binary values; thus, significant correlations with immune cell infiltration were analyzed with the chi-square test or with Fisher’s exact test when fewer observations than 20 were seen.

The prognostic outcome of immune cell infiltration was analyzed with Kaplan–Meier plots and log-rank tests to exclude the null hypothesis of equal prognostic effect for BCSS or RFI based on specific immune cell infiltration in tumor tissues. Effects on BCSS and RFI were calculated based on infiltration of CD169^+^ macrophages alone, TLLSs alone, or dual infiltration of CD169^+^ macrophages and TLLSs (CD169^+^/TLLS). The follow-up data from the TMA cohort enabled a long time to event scale for BCSS and RFI, since all patients had presented with locally advanced or metastatic disease at the time of inclusion into the clinical trial. The time to event scale for BCSS and RFI in the current analysis was censored at 25 years after primary tumor diagnosis. For LNM biopsies, the time to event scale for BCSS and RFI was set to 10 and 25 years after diagnosis in order to focus on both early and long-term prognostic effects, since lymph node metastases were present at primary diagnosis in the majority of patients in the TEX cohort.

Univariable followed by multivariable Cox regression analyses were also performed to estimate hazard ratios (HRs) for recurrence or death from breast cancer according to CD169^+^, TLLS and CD169^+^/TLLS infiltration in PTs and LNMs. The same time to event scale as for the Kaplan–Meier analyses was maintained and the multivariable models accounted for hormone receptor/growth factor expression status (ER, PR, Her2), T_regs_ presence, Ki67 levels, tumor size and age at primary diagnosis. Results were illustrated using forest plots showing HRs with a 95% confidence interval.

### 2.5. Gene Expression Analyses

Gene expression analyses of TLS gene signature [32], B_reg_ signature [33], T_reg_ (*FoxP3*) signature [34] and CD169^+^ TAMs (*SIGLEC1*) were performed using RNA sequencing data from the SCAN-B cohort, following the same procedure as previously described [30,35]. Expression data were extracted as fragments per kilobase per million reads for each case and transformed into a logarithmic scale. Five gene classifiers representing different subtype predictors were used to classify samples into the intrinsic breast cancer subtypes according to the PAM50 gene signature [36]. Prior to analysis, a batch correction was performed via ComBat in order to remove potential bias associated with technical variations. After correction, the data were uploaded unto The Institute for Genomic Research MultiExperiment Viewer (TIGR MeV) version 3.1, and differences in gene expression were determined through hierarchical clustering using median-centered gene correlations where status 1 or above represented upregulated expression and -1 or below represented downregulated expression. RNA sequencing results are presented with heat maps, showing Pearson correlation distance and complete hierarchal clustering linkages.

## 3. Results

### 3.1. CD169^+^ TAMs Associate with TLLSs in PTs

Since SCS CD169^+^ macrophages in lymph nodes have a functional localization surrounding B cell follicles and are associated with a beneficial prognosis in cancer patients, we first set out to investigate the localization pattern of PT-infiltrating CD169^+^ macrophages (CD169^+^ TAMs) in relation to CD20^+^/CD3^+^ B/T cell clusters (referred to as TLLSs) in primary breast cancer tumors (Figure 1B).

Localization patterns were investigated with odds ratios (ORs), as shown in Table 2 and Appendix A. Firstly, CD169^+^ TAMs in PTs were indeed correlated with the presence of TLLSs (*OR* = 3.77, *p* = 0.004) and furthermore showed a trend for T_reg_ infiltration (*OR* = 2.06, *p* = 0.057). CD169^+^ TAMs also correlated with B cells as only marker, unrelated to TLLSs *(OR =* 5.26, *p =* 0.017). In LNMs, CD169^+^ presence (CD169^+^ LNM) was found to also correlate with TLLS presence (*OR* = 4.76, *p* = 0.0001) and T_reg_ infiltration (*OR* = 2.87, *p* = 0.046).

CD169^+^ LNMs further correlated with decreased odds of tumor size above 20 mm (*OR* = 0.42, *p* = 0.041) and increased odds of surviving beyond 5 years (*OR* = 2.20, *p* = 0.045), while CD169 in PTs (CD169^+^ TAMs) correlated with high Ki67 levels (*OR* = 2.33, *p* = 0.021) (Table 2). This was in line with our previously published data using another breast cancer cohort regarding CD169^+^ infiltration in PTs (CD169^+^ TAM) and LNMs (CD169^+^ LNM) [27]. CD169^+^ TAMs also correlated with decreased odds of tumor size above 20 mm (*OR* = 0.47, *p* = 0.019) and decreased odds for expression of ER (*OR* = 0.28, *p* = 0.0001).

Additionally, in PTs, the presence of CD169^+^/TLLS structures showed a trend towards association with decreased odds of surviving the first 5 years (*OR* = 0.29, *p* = 0.053), decreased expression of ER in both PTs and LNMs (*OR_PT-ER_* = 0.26, *p* = 0.042, *OR_LNM-ER_* = 0.10, *p* = 0.051), and high Ki67 levels in PTs (*OR* = 5.43, *p* = 0.018). In sharp contrast, CD169^+^ /TLLS in LNMs was significantly correlated with increased odds of surviving breast cancer the first 5 years (*OR* = 3.51, *p* = 0.005; Appendix A).

To explore the potential univariable role of TLLS infiltration (CD20^+^/CD3^+^; “TLLS”) without CD169^+^ co-localization, further OR analysis was performed between TLLSs and other patient and tumor characteristics (Appendix A). Results indicated that TLLSs in PTs per se (TLLS^+^ PT) was significantly correlated with reduced odds of breast-cancer-related death (*OR* = 0.41, *p* = 0.045) and lower odds for recurrence (*OR* = 0.31, *p* = 0.018) within 5 years after diagnosis. Furthermore, TLLS was significantly correlated with greater odds for infiltration of T_regs_ (FoxP3) (*OR* = 8.54, *p* = 0.001) into PTs. Again, and in contrast, TLLSs in the LMNs per se (TLLS^+^ LNM) was significantly correlated with increased odds of surviving beyond 5 years (*OR* = 2.63, *p* = 0.018), and decreased odds of PR expression in LNM (*OR* = 0.43, *p* = 0.044).

In summary, CD169^+^ macrophages were associated with TLLSs both in PTs and in LNMs, but when present in PTs this was associated to a worse prognosis for the patients, the opposite of what was seen in LNMs.

### 3.2. CD169^+^ TAMs and TLLSs as Prognostic Markers for Breast Cancer Patients

Investigating prognostic impact, Kaplan–Meier plots for each variable (CD169^+^/TLLS; CD169^+^; TLLS) showed unique survival patterns. In general, a worse prognosis was seen with CD169^+^ and TLLS infiltration in PTs, while infiltration in LNMs showed a better prognosis. In PTs, CD169^+^/TLLS co-infiltration was a borderline prognostic marker associated with worse BCSS (*p* = 0.059) (Figure 2A). To estimate if the observed effect was caused by the dual infiltration pattern (CD169^+^/TLLS), or was solely from one type of cell infiltration (CD169^+^ or TLLS), individual Kaplan–Meier plots with corresponding log-rank tests were performed. Both CD169^+^ infiltration (*p* = 0.047) (Figure 2B) and TLLS infiltration (*p* = 0.001) (Figure 2C) in PTs showed evidence for an adverse BCSS. A similar significant inferior outcome regarding RFI was seen only for TLLS infiltration (*p = 0.006*) (Figure 2D–F). Infiltrating B cells as only variable (CD20) however, did not have an impact on survival (BCSS *p* = 0.38; RFI *p* = 0.82).

Conversely, CD169^+^/TLLS presence in LNMs showed significant evidence of improved BCSS in the first 10 years (*p* = 0.016) (Figure 2G). Individually, CD169^+^ LNM presence remained significant (*p* = 0.023), while TLLSs showed weaker evidence (*p* = 0.083) for a beneficial prognostic effect on BCSS (Figure 2H,I). For RFI, no statistically significant correlations were observed for any type of immune cell infiltration; the survival curves, however, did trend towards longer RFI upon TLLS infiltration alone or CD169^+^/TLLS co-presence, suggesting a potential beneficial prognostic effect (Figure 2J–L). The beneficial prognostic effects for CD169^+^ and TLLS in LNMs, however, were lost in the long-term 25-year follow-up (Appendix A).

Because the prognostic effect was opposite that based on tumor localization (PT vs. LNM), we further investigated the prognostic impact for patient matched biopsies. Interestingly, the results suggested that CD169^+^ TAM infiltration in PTs was relevant as a prognostic factor only if CD169^+^ macrophages in LNMs were absent, and vice versa. The same finding was true for TLLSs (Appendix A).

Hence, the observed opposite prognostic effects in PTs and LNMs were seen both for individual (CD169^+^ or TLLS) and dual infiltration patterns (CD169^+^/TLLS).

### 3.3. T_reg_ Infiltration Impacts the Prognostic Effect of CD169^+^ TAMs

In previous research using the TMA cohort, T_reg_ infiltration in PTs was found to be an independent prognostic factor for decreased BCSS, but the prognostic effect was lost in LNMs [31]. In line with this, in the present study we show that CD169^+^ TAMs trended towards an association with infiltration of T_regs_ (OR = 2.06, *p* = 0.057), and that TLLSs correlated with T_regs_ (*OR* = 8.54, *p* = 0.001) (Table 2 and Appendix A). Interestingly, CD169^+^/TLLS dual infiltration in PTs (CD169^+^/TLLS PT) was significantly associated with the opposite, meaning a decreased presence of T_regs_ (FoxP3^+^) in PTs (*OR* = 0.59, *p* = 0.007) (Appendix A). To interpret this, we further investigated the impact of CD169^+^ TAMs and TLLS infiltration based on a FoxP3^+^ (T_reg_) strata in PT biopsies only. To our surprise, we found that PTs lacking T_reg_ infiltration also always lacked CD169^+^/TLLS dual infiltration, implying that co-infiltration of CD169^+^ TAMs and TLLSs is necessary for the presence of T_regs_, and vice versa. Therefore, Kaplan–Meier plots with corresponding log-rank tests could not be performed with a FoxP3^+^ strata for CD169^+^/TLLS and TLLs. Importantly, however, individual analysis of CD169^+^ infiltration alone (CD169^+^ TAM) associated with shortened RFI (*P_RFI_* = 0.001) and a trend towards association with shortened BCSS (*P_BCSS_* = 0.055) only in the absence of FoxP3^+^ T_regs_ in PTs (Figure 3A,B). Hence, the prognostic effects of CD169^+^ TAMs alone for BCSS and RFI were completely lost in the FoxP3^+^ T_regs_ positive patient strata (Figure 3C,D).

Altogether this suggests that co-infiltration of CD169^+^/TLLS may be necessary for the presence of T_regs_, and that the presence of CD169^+^ TAMs alone may only have a prognostic impact in breast tumors lacking T_regs_.

### 3.4. CD169^+^ TAMs and TLLSs Show Unique Independent Prognostic Effects

Multivariable Cox regression analyses were done to compare the effects from each type of cell infiltration biomarker adjusted for several potential confounders. Included confounders taken into account were: age, nodal status, tumor size, Ki67, receptor status (ER, PR, HER2), T_regs_ presence, TLLS (CD20^+^/CD3^+^) presence and CD169^+^ macrophage presence (Appendix A). The prognostic impact was calculated with HR with a 25-year timeline in PT samples and a 10-year timeline in LNM samples for BCSS and RFI. For BCSS, after multivariable adjustments, dual infiltration of CD169^+^/TLLS in PTs showed an independent HR value correlating to a worse prognosis (*HR* = 2.88, 95%CI: (1.33–6.2), *p* = 0.007), hence even stronger than the univariable effect (*HR* = 1.90, 95%CI: (0.97–3.75), *p* = 0.063) (Appendix A). In contrast, in LNMs, CD169^+^/TLLS dual infiltration was correlated with improved survival in univariable analysis only (*HR* = 0.54, 95%CI: (0.33–0.90), *p* = 0.017). In the multivariable analysis, CD169^+^/TLLS dual infiltration in LNMs showed a similar trend but with weaker statistical evidence (*HR* = 0.45, 95%CI: (0.20–1.02), *p* = 0.057) (Appendix A). CD169^+^ and TLLS infiltration were next analyzed separately. Multivariable adjustments further strengthened TLLSs as an independent prognostic marker in PTs (*HR _TLLS_* = 1.73, 95%CI (1.03–2.93), *p* = 0.040), while evidence for CD169^+^ TAMs as a prognostic factor in PTs decreased (*HR _CD169_* = 1.07, 95%CI: (0.67–1.71), *p* = 0.77) compared to their univariable effects (*HR _TLLS_* = 2.14, 95%CI: (1.38–3.35), *p* = 0.001; *HR _CD169_* = 1.43, 95%CI (1.002–2.04), *p* = 0.049) (Appendix A). In contrast to PTs, multivariable analyses of CD169^+^ LNMs showed both stronger HRs and stronger correlation to decreased risk of death from breast cancer, while evidence for TLLS infiltration being an independent prognostic factor decreased (*HR _CD169_* = 0.48, 95%CI: (0.23–0.99), *p* = 0.046; *HR _TLLS_* = 0.72, 95%CI: (0.40–1.31), *p* = 0.28) compared to their univariable effects (*HR _CD169_* = 0.59, 95%CI: (0.37–0.93), *p* = 0.025; *HR _TLLS_* = 0.66, 95%CI: (0.41–1.06), *p* = 0.085) (Appendix A).

Regarding the independent prognostic impact on RFI, it was clear that the prognostic effect was not as important as for BCSS. Multivariable analysis for CD169^+^ TAMs and TLLS infiltration, respectively, showed non-significant HR correlations in both PTs and LNMs. However, CD169^+^/TLLS dual infiltration in PTs was significantly correlated with higher HRs of recurrence (*HR* = 2.15, 95%CI: (1.06–4.38), *p* = 0.035) (Appendix A).

Our data thus showed that CD169^+^/TLLS dual infiltration in PTs of advanced breast cancer patients was an independent prognostic marker with regards to both BCSS and RFI.

### 3.5. CD169^+^ TAMs Associate with Both Mature TLS and B_reg_ Gene Signatures

To investigate whether the TLLSs associating with CD169^+^ TAMs in PTs were functional tertiary lymphoid follicles, and to confirm the association between CD169^+^ TAMs and TLLSs in breast cancer, gene signatures of mature tertiary lymphoid structure (TLS) [32] from bulk RNAseq from 8164 patients of the SCAN-B cohort were investigated, allowing analysis in a larger, contemporary and representative cohort. Initial results (Figure 4A), showed that CD169^+^ expression (gene name, *SIGLEC1*) indeed correlated with a mature TLS gene signature for a specific cluster of patients, indicating functional TLS formations. This specific cluster also had upregulated levels of the gene *MS4A1*, encoding CD20. For all clusters, the molecular subtype, although being dominant for more aggressive subtypes (luminal B, HER2-enriched, basal), was not restricted to one subtype cluster. This implies that TLS formation can occur in all subtypes of breast cancer.

To further investigate the possible immunological role of the CD169^+^ TAMs in the PT, we analyzed other immunoregulatory gene signatures [33,34]. CD169^+^ TAM upregulated clusters correlated with the B_regs_ gene signature (Figure 4B). Notably, a cluster of patients with CD169^+^ TAMs further associated with TLS, B_reg_ and T_reg_ gene signatures (Figure 4C), indicating an immunosuppressive function for these TLSs. More importantly, gene signatures for TLS formation were not seen in patient clusters that lacked CD169^+^ TAMs, and the same observations were true for T_reg_ and B_reg_ gene signatures. Lastly, CD169^+^ TAM positive clusters also associated with genes important for CD169^+^ macrophage biology, such as *CD163*, *CSF-1*, *LTA* and *LTB* (Figure 4D). Complete heatmaps with the highlighted clusters are presented in Appendix A.

In summary, this implies that CD169^+^ TAMs are closely connected to mature TLS formation, and T_reg_ and B_reg_ infiltration, in PTs of breast cancer patients.

## 4. Discussion

In secondary lymphoid organs (SLOs), the CD169^+^ SCS lymph node macrophages surround B cell follicles to aid in antigen delivery and to regulate immune responses [19]. Our initial hypothesis was therefore that CD169^+^ TAMs would associate with TLLSs in PTs, just like CD169^+^ SCS macrophages do in SLOs, hence aiding or regulating immune activity. CD169 is primarily expressed on activated monocytes and macrophages, with occasional expression on T cells and mature dendritic cells (DCs) [37,38]. Here, co-stainings with CD3 ruled out T cells expressing CD169; however, we cannot exclude that follicular DCs could potentially express microvesicles containing CD169 derived from macrophages [39]. Our original published data, which were supported by this present study, showed that CD169^+^ TAMs infiltrating PTs were indeed associated with a worse prognosis for breast cancer patients [27]. We here show that CD169^+^ TAMs infiltrating PTs actually do associate with TLLSs in PTs, similar to the CD169^+^ SCS resident lymph node macrophages and B-cell follicles in SLOs. We show that CD169^+^ macrophages are in close spatial association with TLLSs in PTs, and surprisingly also with presence of T_regs_. In spleen, CD169^+^ macrophages are dependent on lymphotoxin α1ß1 generated from B-cell follicles [40,41]. An explanation for the co-localization of CD169^+^ TAMs and TLLSs in breast tumors could therefore be a local secretion of lymphotoxin α1ß1 in tumors with TLSs. Indeed, our bulk RNA-sequencing data from the SCAN-B cohort showed that upregulation of CD169/*SIGLEC1* also correlates with *LTA1* and *LTB1* upregulation.

Both CD169^+^ TAMs and TLLSs in PTs were clearly associated with a worse prognosis, in contrast to CD169^+^ macrophages present in LNMs, which had a beneficial effect on prognosis. This finding is in disagreement with previous literature where TLS presence in primary breast tumors was a positive prognostic factor [42,43], as a meta-analysis has shown TLS-presence to generally be associated with a beneficial prognosis in breast cancer [43]. However, in previous studies, the beneficial effect of TLSs as a prognostic marker in PTs was highly dependent on breast cancer molecular subtypes (HER2 amplified [44] or TNBC [45]). These data should be put into relation with the TMA cohort used in the present study, which was comprised mostly of luminal tumors (*n* = 143) and very few HER2^+^ (*n* = 9) and TNBC (*n* = 24) tumors. All the patients used in the TMA cohort, furthermore, had developed metastatic disease and therefore had a poor prognosis. This could indicate differential impact of TLS depending on the molecular subtype or due to the advanced stage. One observation supporting this was made by Figenschau et al. [46], who verified that tumors with a higher level of tumor infiltrating immune cells correlated with intra-tumoral TLS formation, higher tumor grade and a higher degree of inflammation, thus leading to worse prognosis. We also found that presence of CD169^+^ LNM and CD169^+^ PT showed lower odds of having a large tumor size. This is surprising given the worse prognosis seen for CD169^+^ TAMs in PTs, although when adjusting for multiple variables the prognostic effect of CD169^+^ PT was lost. Nonetheless, these findings could indicate a more aggressive behavior and hence microenvironment of primary tumors with CD169^+^ TAMs already at a low tumor size in this cohort with advanced breast cancer patients. Lastly, a general difficulty when investigating TLSs in lymph node metastases is the lobular structure of the SLO, since secondary lymphoid follicles may be difficult to separate from TLSs in the sectioned lymph node metastases. How to discriminate TLS from secondary lymphoid follicles will be an important issue to solve for the TLS immune oncology field in the future.

Using RNA sequencing data from the large SCAN-B breast cancer cohort, we showed that *SIGLEC1* (CD169) expression in primary breast tumors clustered with the expression of functional TLS signatures, indicating that CD169^+^ TAMs actually do associate with mature tertiary lymphoid follicles also in primary tumors. A fraction of these were enriched for B_reg_ and T_reg_ signatures, thus possibly inducing immunosuppression and adverse prognostic effects in breast cancer patients. This would be supported by a recent study showing that the presence of TLSs with B_reg_ and T_reg_ infiltration was associated with a worse prognosis in primary breast tumors from invasive ductal carcinoma (IDC) and ductal carcinoma in situ (DCIS) [47]. In the present study, we evaluated this hypothesis using RNA sequencing data. In fact, there was a clear subcluster of patients with high transcript levels for CD169^+^, TLS gene signatures, and B_reg_ and T_reg_ signatures, possibly leading to immunosuppression and hence a worse prognosis. Furthermore, TLS and B_reg_ gene signature transcripts were only present for subsets of patients with a higher infiltration of CD169^+^ TAMs, implying that CD169^+^ TAMs, TLS and B_regs_ may interact. On the other hand, patients with both CD169 and TLS signatures, presenting with lower expression of the B_regs_ signature, were numerous in comparison. The significance of these findings in relation to outcome will be interesting to evaluate and will be a future goal.

Another important result from our study was the correlation between TLS, CD169^+^ TAMs and FoxP3^+^ T_regs_. We found a strong correlation between FoxP3^+^/CD4^+^ T_reg_ signatures and CD169^+^ expression (*SIGLEC1*) in the SCAN-B cohort, and the T_reg_ gene signature also correlated with TLS and B_reg_ gene signatures. Indeed, based on our IHC results using the TMA cohort, TLS infiltration was almost exclusively present when T_regs_ were also present in PTs. In fact, several papers have shown that the presence of T_regs_ is associated with a worse prognosis in breast cancer [31,48,49]. As the main purpose of T_regs_ is to suppress lymphocytes, it is likely that FoxP3^+^ T_reg_ infiltration is a natural feedback response following high levels of TLLS formation, suppressing these lymphocytes. A previous in vivo study investigating the correlation between T_regs_ and TLS in lung cancer showed that T_regs_ actively suppress the anti-tumor response from TLSs. In the same model, T_regs_ depletion reversed this effect and led to T-cell expansion starting at the TLS sites and promoting tumor destruction [50]. It was also recently shown that tumors affect local lymph node immune tolerance epigenetically via type I IFNs, eventually promoting distant metastasis facilitated by tumor-antigen specific T_regs_ in a malignant melanoma model [51]. As lymph node CD169^+^ macrophages are known type I IFN producers [12], the correlation seen here between CD169^+^ macrophages and T_regs_ could possibly be involved in this epigenetic reprograming [51]. Even more interesting is that T_regs_ infiltration is associated with a higher risk of death and relapse, especially for ER positive breast cancer patients [49], which is in line with our TMA cohort that was predominantly of luminal subtype. The presence of T_regs_ likely inhibits reactivation of T-cell responses; thus, despite the presence of TLLSs, the immune system is unable to counteract tumor growth. A final interesting finding regarding T_regs_ from the present study was that co-infiltration of CD169^+^ TAMs/TLLS is necessary for the presence of T_regs_, and that CD169^+^ TAMs alone only had a prognostic impact in tumors lacking T_regs_. This most probably means that T_regs_ have a dominant functional role over CD169^+^ TAMs. CD169^+^ TAMs may also play another important role here in relation to T_regs_ infiltration, as it has been shown that CD169^+^ macrophages can upregulate CCL22 upon interaction with apoptotic cells, leading to FoxP3^+^ T_reg_ recruitment via a CCL22/CCR4-mediated chemotaxis gradient [52]. The same mechanism might apply within breast cancer patients with high tumor grade, with an abundance of apoptotic cells or necrotic cells, thus recruiting CD169^+^ TAMs which release CCL22 and recruit T_regs_.

Lastly, the results we obtained, where the presence of CD169^+^ TAMs and TLLSs were correlated with prognosis only for patients who had immune cell infiltration in either PT or in LNM, but not for both, are interesting. Subsequently, patients who had CD169^+^ TAMs or TLLS infiltration in both PT and LNM had similar survival compared to patients that lacked the same in both PT and LNM. This result either implies that the harmful prognostic effect of immune cell infiltration seen in PTs cancels out the beneficial prognostic effect seen in LNMs, or that infiltration of CD169^+^ TAMs into PTs is concurrent with the depletion of CD169^+^ macrophages from the LNMs in breast cancer patients with a higher tumor grade, therefore giving rise to worse prognosis within our cohort that only includes patients with advanced breast cancer.

## 5. Conclusions

In conclusion, CD169^+^ macrophages present in breast cancer PTs and LNMs correlate to the presence of TLS, T_reg_ and B_reg_ signatures for a subset of patients. This was associated to worse survival when present in PTs, while conferring a better prognosis when present in LNMs (Figure 5). We propose that attraction and polarization of CD169^+^ TAMs occur in tumors where formation of TLSs occurs, and that these inflamed environments cause enrichment of immunosuppressive regulatory lymphocytes, T_regs_ and B_regs_, thus fueling even more immunosuppressive environments and breast tumor progression.

## Figures and Tables

**Figure 1 cancers-15-01262-f001:**
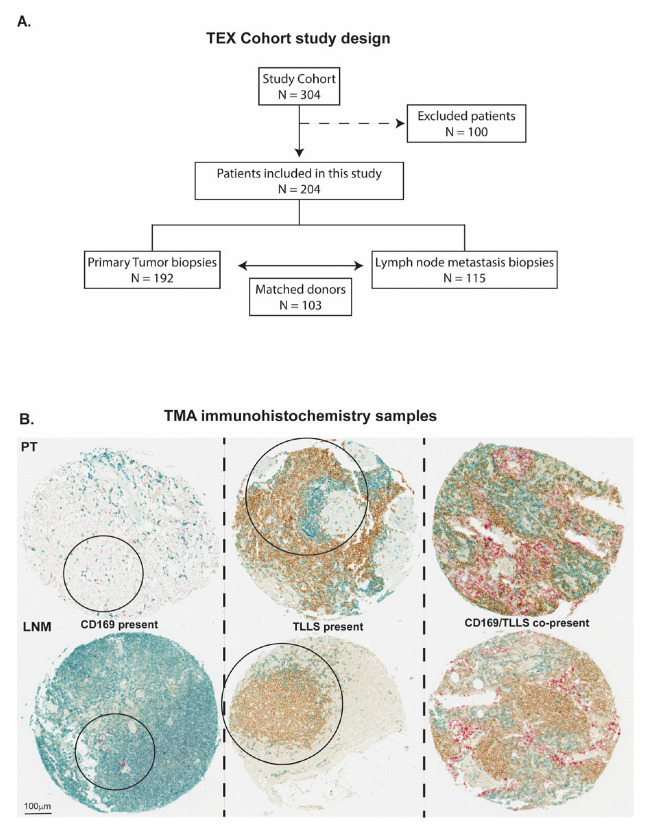
The TEX cohort study design flow chart and immunohistochemical staining examples from the cohort. (**A**) 204 patients were included from the retrospective TEX cohort, giving altogether 192 primary tumor and 115 lymph node metastasis biopsy samples; excluded patients lacked or had missing biopsy cores. (**B**) Tissue microarray (TMA) sections and immunohistochemistry using the three markers CD169 (red), CD20 (brown) and CD3 (blue). The staining panel allowed for identification of three types of cell infiltration/presence patterns, which are highlighted in circles, in primary tumors (PTs) and lymph node metastases (LNMs); CD169^+^ macrophages only (CD169^+^), tertiary lymphoid-like structures only (TLLS), and CD169^+^ macrophages together with TLLS (CD169^+^/TLLS).

**Figure 2 cancers-15-01262-f002:**
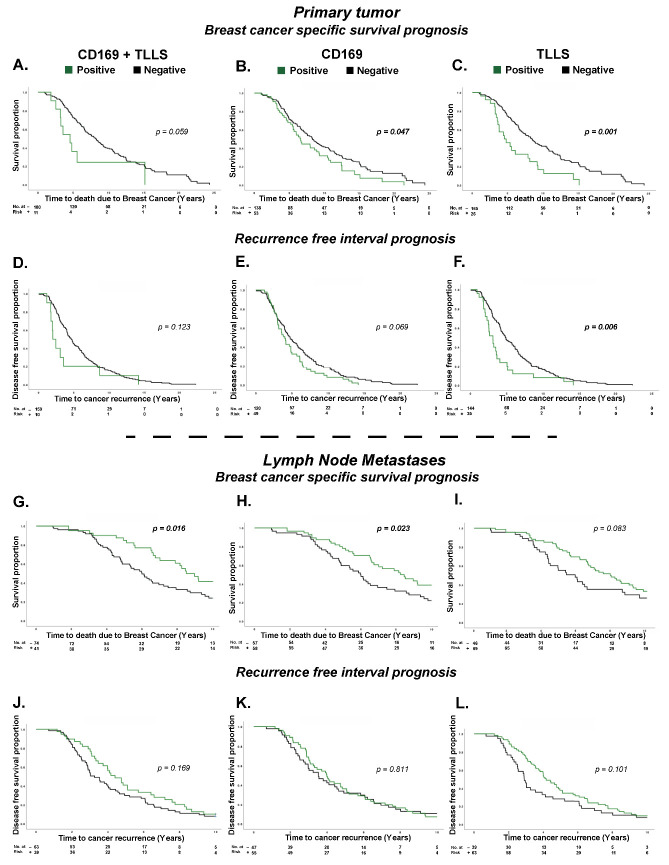
Kaplan–Meier survival plots investigating differences in 25-year breast cancer specific survival (BCSS) and recurrence free interval (RFI) for specific immune cell populations infiltrating tumors. P values by the log-rank test are highlighted in bold when significant. In panels (**A**–**F**), the impact of immune cell infiltration for CD169^+^ TAMs, TLLSs and CD169^+^ TAMS/TLLS was investigated as prognostic markers for BCSS and RFI in primary tumors (PTs). In panels (**G**–**L**), the impact of CD169^+^ TAMs, TLLSs and CD169^+^ TAMS/TLLS on BCSS and RFI was investigated in lymph node metastases (LNMs). Green lines indicate PTs and LNMs with CD169^+^ TAMs, TLLSs or CD169^+^ TAMS/TLLS infiltration, and black lines indicate the absence of CD169^+^ TAMs, TLLSs or CD169^+^ TAMS/TLLS.

**Figure 3 cancers-15-01262-f003:**
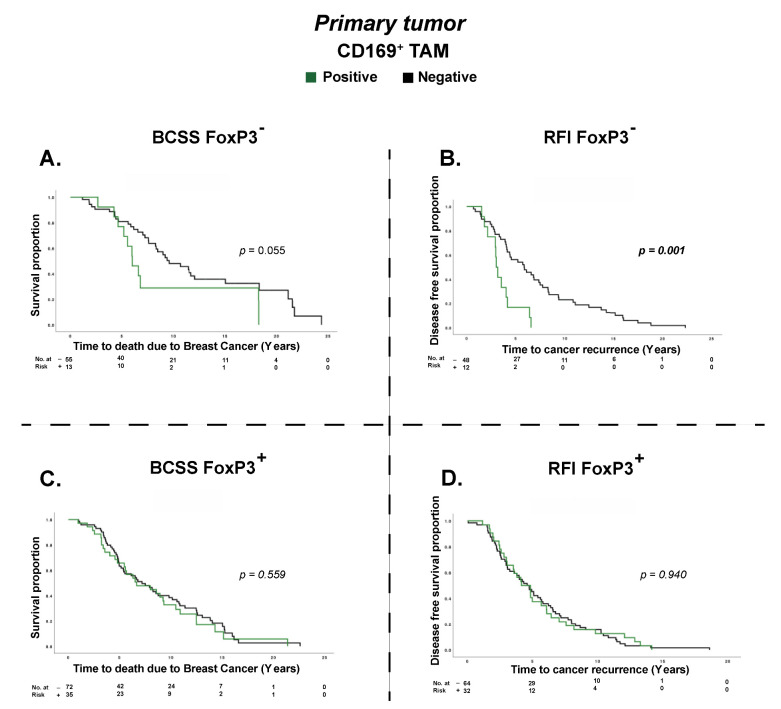
Kaplan–Meier survival plots with a FoxP3 strata on BCSS and RFI in PTs only. *p* values by the log-rank test and highlighted in bold when significant. Panels (**A**,**B**) show correlations for BCCS and RFI with CD169^+^ TAMs in FoxP3 negative cases, while panels (**C**,**D**) show correlations for BCCS and RFI with CD169+ TAMs in FoxP3 positive tumors. For all panels, green lines indicate PTs with CD169^+^ TAMs and black lines indicate patients with the absence of CD169^+^ TAMs.

**Figure 4 cancers-15-01262-f004:**
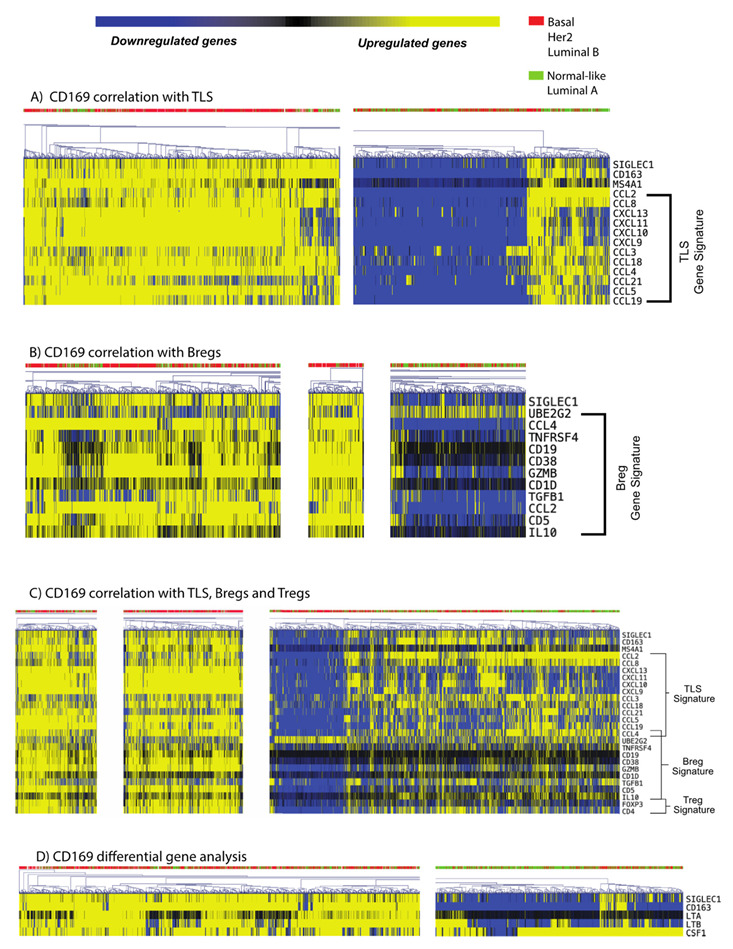
Heat map associations between SIGLEC1 (CD169) and gene signatures for (**A**) tertiary lymphoid structures (TLSs), (**B**) Bregs and (**C**) Tregs. Patients are characterized based on their molecular subtype of breast cancer, aggressive subtypes in red (basal-like, HER2^+^ and luminal B), and luminal A or normal-like subtypes in green. Upregulated genes are shown in yellow while downregulated genes are shown in blue. The threshold for upregulation/downregulation was set at 1/-1 based on median-centered genes relations. The highlighted areas represent clusters with positive cell infiltration, taken from full-scale analysis shown in Appendix A. (**D**) SIGLEC1 (CD169) was correlated with the CD169^+^ macrophage differentiation markers CD163, LTA, LTB and CSF1.

**Figure 5 cancers-15-01262-f005:**
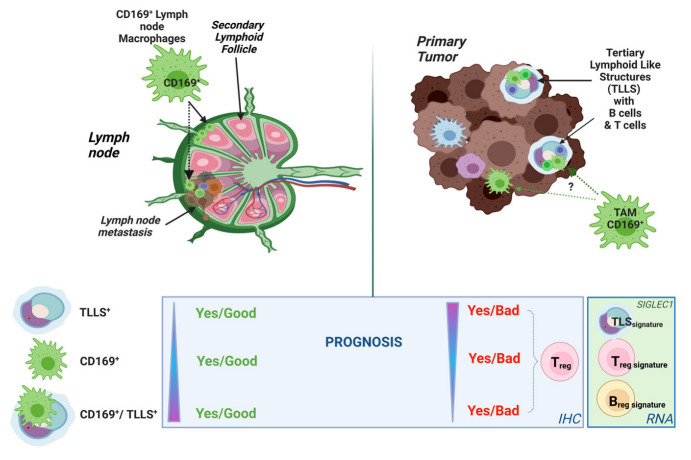
Schematic summary of results. Made in biorender.com. https://biorender.com (accessed on 9 February 2023).

**Table 1 cancers-15-01262-t001:** Patient characteristics and clinicopathological features of patients included in the TEX study [28].

Patient Characteristics		No. of Patients	Percent (%)
Age	>50	92	45.1
	<50	112	54.9
Tumor size (T)	T1 (0–20 mm)	83	40.7
	T2 (20–50 mm)	95	46.6
	T3 (>50 mm)	16	7.8
	T4 (Invasion)	9	4.4
	Missing	1	0.5
Regional lymph nodes (N)	N0	65	31.9
	N1	124	60.8
	N2	9	4.4
	N3	2	1.0
	Missing	4	2.0
Metastasis (M)	M0	185	90.7
	M1	18	8.8
	Missing	1	0.5
PT receptor status			
ER	Neg	36	17.6
	Pos	152	92.2
	Missing	16	7.8
PR	Neg	80	39.2
	Pos	107	52.5
	Missing	17	8.3
HER2	Neg	172	84.3
	Pos	17	8.3
	Missing	15	7.4
LNM receptor status			
ER	Neg	28	13.7
	Pos	74	36.3
	Missing	102	50.0
PR	Neg	64	31.4
	Pos	38	18.6
	Missing	102	50.0
HER2	Neg	77	37.7
	Pos	13	6.4
	Missing	114	55.9
Adjuvant therapy given			
Chemotherapy	No	106	52.0
	Yes	98	48.0
Endocrine	No	92	45.1
	Yes	112	54.9
Radiotherapy	No	55	27.0
	Yes	149	73.0

Abbreviations: ER = estrogen receptor; PR = progesterone receptor; HER2 = human epidermal growth factor receptor 2.

**Table 2 cancers-15-01262-t002:** Odds ratio table comparing CD169^+^ macrophage infiltration in PTs and LNMs with tumor and metastasis clinicopathological features as well as other immune cell infiltration.

Clinicopathological Features		CD169^+^ PT				CD169^+^ LNM			
		OR	95% CI	*p*-Value ^a^	*n*	OR	95% CI	*p*-Value ^a^	*n*
Age	>50				85	1			48
	<50	0.96	0.51–1.81	0.90	106	0.89	0.43–1.88	0.77	67
Overall Survival	>5 years	1			67	1			42
	<5 years	1.21	0.61–2.36	0.59	124	2.20	1.01–4.79	0.045	73
Relapse free interval	>5 years	1			108	1			77
	<5 years	0.61	0.31–1.22	0.16	73	1.37	0.61–3.06	0.44	35
Tumor size	T1	1			80	1			34
	>T1	0.47	0.24–0.89	0.019	110	0.42	0.19–0.97	0.041	80
Ki67^+^ PT	Neg	1			24	1			37
	Pos	2.33	1.183–4.600	0.021	24	1.26	0.535–2.989	0.67 ^b^	16
Ki67^+^ LNM	Neg	1			14	1			31
	Pos	1.47	0.550–3.923	0.45^b^	9	0.69	0.285–1.662	0.51^b^	12
ER PT	Neg	1			36	1			21
	Pos	0.28	0.13–0.60	0.001	147	1.43	0.55–3.75	0.63	85
ER LNM	Neg	1			24	1			28
	Pos	0.76	0.26–2.28	0.77	69	0.53	0.22–1.29	0.18	70
PR PT	Neg	1			79	1			42
	Pos	0.59	0.31–1.12	0.11	103	1.18	0.54–2.59	0.68	61
PR LNM	Neg	1			57	1			60
	Pos	0.34	0.11–1.01	0.053	37	0.68	0.30–1.56	0.36	37
HER2 PT	Neg	1			166	1			95
	Pos	1.77	0.61–5.17	0.37^b^	16	0.65	0.17–2.46	0.74 ^b^	10
HER2 LNM	Neg	1			69	1			77
	Pos	0.99	0.24–4.06	1^b^	13	1.41	0.41–4.76	0.76 ^b^	12
Cell infiltration association									
FoxP3 PT	Neg	1			68	1			40
	Pos	2.06	0.99–4.26	0.057	107	0.76	0.34–1.72	0.51	56
FoxP3 LNM	Neg	1			24	1			26
	Pos	0.70	0.23–2.13	0.57	49	2.87	0.99–8.27	0.046	54
TLLS PT	Neg	1			165	1			91
	Pos	3.77	1.61–8.82	0.004	26	2.04	0.58–7.27	0.36 ^b^	12
TLLS LNM	Neg	1			39	1			46
	Pos	1.02	0.40–2.62	0.97	64	4.76	2.12–10.71	0.0001	69

Abbreviations: PT = primary tumor; LNM = lymph node metastases; OR = odds ratio; 95% CI = 95% confidence interval; *n* = number of patients; ER = estrogen receptor; PR = progesterone receptor; HER2 = human epidermal growth factor receptor 2; ^a^ = Fisher exact test unless otherwise stated, two-tailed; ^b^ = Pearson’s chi square test, two-tailed.

## Data Availability

Gene expression data are available at Mendeley Data as a publicly accessible dataset (https://data.mendeley.com/datasets/yzxtxn4nmd, accessed on 17 May 2022) (https://doi.org/10.17632/yzxtxn4nmd.1) [49].

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
