# Peer review of "CD169+ Macrophages in Primary Breast Tumors Associate with Tertiary Lymphoid Structures, Tregs and a Worse Prognosis for Patients with Advanced Breast Cancer"

_cancers, 2023, doi:10.3390/cancers15041262_

Round 1

Reviewer 1 Report

Authors build a co-relative analysis of patient TLS with survival outcome and relation with myeloid cells and Tregs in breast cancer patients.

While having a mechanism of action can add significant benefit but given this is a clinical paper and has a significant value to biomarker identification, it is very timely to get published. Having this in mind, I encourage the editor to accept and expecdite the publication.

Author Response

Reviewer 1. Authors build a co-relative analysis of patient TLS with survival outcome and relation with myeloid cells and Tregs in breast cancer patients.

While having a mechanism of action can add significant benefit but given this is a clinical paper and has a significant value to biomarker identification, it is very timely to get published. Having this in mind, I encourage the editor to accept and expecdite the publication.

We sincerely thank reviewer 1 for these encouraging comments. We agree that the findings are important to publish right now, especially since the field of TLS and TAMs are in focus, with potentially important interpretations to be made from our findings.

Reviewer 2 Report

CD169+ Macrophages in Primary Breast Tumors Associate with Tertiary Lymphoid Structures, Tregs and a Worse Prognosis for Patients with Advanced Breast Cancer

Oscar Briem 1, Eva Källberg 1, Siker Kimbung 2, Srinivas Veerla 2, Jenny Stenström 3, Thomas Hatschek 4, Catharina 5 Hagerling 3, Ingrid Hedenfalk 2 and Karin Leandersson 1,*

1 Cancer Immunology, Department of Translational Medicine, 214 28 Malmö, Lund University, Sweden 7

2 Division of Oncology, Department of Clinical Sciences; Lund University, 223 81 Lund, Sweden 8

3 Division of Clinical Genetics, Department of Laboratory Medicine Lund; Lund University, 221 84 Lund, 9 Sweden 10

4 Department of Oncology and Pathology, Karolinska Institutet, 171 77 Solna, Sweden 11

* Correspondence: Karin.Leandersson@med.lu.se; Tel.: +4640391134.

Briem et al. have submitted a good manuscript investigating the role of CD169+ TAMs in TLLS in breast cancer. CD169+ TAMs correlate with TLS, Treg and Breg signatures in some subsets of metastatic breast cancer patients. In addition, CD169+ and TLLS  in the tumor associated with a worse prognosis, in contrast to previously published work in breast cancer. However, the advanced stage of the breast cancer patients in this study could explain the differences between this study and previous studies and warrants further research towards the role of TLS is advanced stage cancers. Overall, this study both confirms previously published data on CD169+ TAMs and TLS in LNM and adds novel data on the correlations of CD169+, TLLS and prognosis of breast cancer patients. Therefore, the reviewer suggests to accept this manuscript for publication with minor comments.

Minor comments:

·       Due to the high level of statistical comparisons made, the manuscript would benefit from a schematic overview of the results

·       The authors have only used CD169 (SIGLEC1) to identify CD169+ TAMs. The authors should comment on potential CD169 expression on other cell types influencing these results.

·       Could the authors elaborate in the discussion on the correlation of CD169+ TAMs on the lower odds of having a tumor size above 20 mm.

Author Response

Reviewer 2. Briem et al. have submitted a good manuscript investigating the role of CD169+ TAMs in TLLS in breast cancer. CD169+ TAMs correlate with TLS, Treg and Breg signatures in some subsets of metastatic breast cancer patients. In addition, CD169and TLLS  in the tumor associated with a worse prognosis, in contrast to previously published work in breast cancer. However, the advanced stage of the breast cancer patients in this study could explain the differences between this study and previous studies and warrants further research towards the role of TLS is advanced stage cancers. Overall, this study both confirms previously published data on CD169+TAMs and TLS in LNM and adds novel data on the correlations of CD169+, TLLS and prognosis of breast cancer patients. Therefore, the reviewer suggests to accept this manuscript for publication with minor comments.

We thank reviewer 2 for the nice summary of the field and for putting our data into perspective.

Minor comments:

  • Due to the high level of statistical comparisons made, the manuscript would benefit from a schematic overview of the results

Thank you for this suggestion – we have now made a schematic summary and added as Figure 5.

  • The authors have only used CD169 (SIGLEC1) to identify CD169TAMs. The authors should comment on potential CD169 expression on other cell types influencing these results. 

We appreciate this comment and have now added a sentence on this in the discussion on row 406-410 discussing that T cells can be excluded since we co-stained with CD3, that mature DCs are rare in breast cancer according but that we cannot exclude that follicular DCs could potentially express microvesicles derived from macrophages containing CD169 (Chen et al., 2022).

  • Could the authors elaborate in the discussion on the correlation of CD169+TAMs on the lower odds of having a tumor size above 20 mm.

Thank you for lifting this, we have now added a sentence in the discussion regarding this on row 436-441.

Reviewer 3 Report

The manuscript by Brien and colleagues aims to dissect the reason behind the differential outcome of advanced breast cancer patients when considering CD169-positive macrophages in primary tumors vs lymph node (LN) metastases. Their approach is based on correlative association between histopathological annotations and immunohistochemical staining of different markers. Although the results are certainly interesting, well-presented and also put into a broader context (former research from the authors as well as evidence from relevant scientific publications), it is a pity that the study remains purely correlative and does not dive into the biology underlying these clinical aspects. Even more so when taking a look at the output of the authors, who seem to have often included more functional and biology-oriented assays in their work.

In order for the manuscript to be ready for submission, I would like the author to clarify/speculate about a series of questions:

- How do authors define a TLLS in the context of a lymphoid organ such as the LN?

- Given the relevance of the lympho(vascular) system in mediating the trafficking of leucocytes, is there any related features (e.g. LVI) that could have been used in their multivariable analysis?

- I was intrigued by the regulatory B cell circuitry. First, are Bregs physically part of a TLS? If so, is their presence is associated with specific developmental stages of a TL(L)S? Accordingly, is the extent of B-cells linked to prognosis?

- Authors stress that CD169 TAMs are associated with mature TLS formation. Does it imply that it is possible to find non-mature TLS (or other different stages) without CD169 TAM? Is it known which event is causal, CD169 or TLS?

- In the discussion, authors bring together TILs and TLS. It seems clear that there is both a spatial and temporal dependence in the cascade of events, as high TILs count is generally a favourable features for patients, while the data presented by the authors seem to indicate that CD169 TAMs and TLLS are detrimental. Do authors have TILs count for the patients in their cohort as an additional variable for their analysis? Also, does it matter how Tregs and TLLS are spatially distributed and communicate with each other? As the count is made through pure H/E, while a TLS would require a series of staining, how do the author reason about TILs and TLS?

- The relationship between CD169 in PT vs LN met and its prognostic value is really interesting. It made wonder whether this could actually be due to a phenomenon recently described by Reticker-Flynn et al (Cell, 2022). In this work, authors provide data supporting the hypothesis that tumor cell infiltration to LN is not directly required for metastasis but rather for the generation of a local immune tolerance (through chronic IFN-gamma and epigenetic rewiring), that can be later spread systemically by Tregs. Eventually, this mechanism enables the outgrowth of distant metastases.  Can authors speculate about it?

Text format&typos:

- line 138 should read 800 um and not mm

- line 452 should read starting at the TLS sites  and not starting the TLS sites

Author Response

Reviewer 3. The manuscript by Brien and colleagues aims to dissect the reason behind the differential outcome of advanced breast cancer patients when considering CD169-positive macrophages in primary tumors vs lymph node (LN) metastases. Their approach is based on correlative association between histopathological annotations and immunohistochemical staining of different markers. Although the results are certainly interesting, well-presented and also put into a broader context (former research from the authors as well as evidence from relevant scientific publications), it is a pity that the study remains purely correlative and does not dive into the biology underlying these clinical aspects. Even more so when taking a look at the output of the authors, who seem to have often included more functional and biology-oriented assays in their work.

In order for the manuscript to be ready for submission, I would like the author to clarify/speculate about a series of questions:

We thank reviewer 3 for these comments and assure that we aim to eventually understand the mechanisms of CD169 TAMs. However, CD169+ TAMs are much more difficult to study than other TAMs for various reasons. To generate accurate hypotheses, we first need to interpret their spatial correlation with other immune cells in cancer cohorts (here TLLS). In this study this is what we focused on and we sincerely believe that this has been achieved with this study.

- How do authors define a TLLS in the context of a lymphoid organ such as the LN?

This is an important comment that we believe the TLS field in general needs to lift for discussion more. In this manuscript, we have only studied TLLS in cores from the metastastic lesions in lymph node (LNM) since this is what was used for the TMA construction. Indeed, some LNM TLS may be in close association with metastasis and not inside the metastasis. Also a secondary lymphoid organ with lobules, may be difficult to interpret using IHC, since sectioning may accidentally include a lobe with primary lymphoid follicles even when aiming to investigate a metastasis. We have therefore added a sentence on this important issue in the discussion. Row 441-446.

- Given the relevance of the lympho(vascular) system in mediating the trafficking of leucocytes, is there any related features (e.g. LVI) that could have been used in their multivariable analysis?

This is an interesting issue, but we have unfortunately not analyzed LVI relevance in this particular breast cancer cohort. Since LVI criteria have previously been shown to be prognostic in early breast cancer (our cohort is advanced breast cancer), we hope that reviewer 3 accepts that we do not include it in this study.

- I was intrigued by the regulatory B cell circuitry. First, are Bregs physically part of a TLS? If so, is their presence is associated with specific developmental stages of a TL(L)S? Accordingly, is the extent of B-cells linked to prognosis?

The current knowledge in this field is still developing. In the presence of follicular T-cells producing IL-21, there is evidence of IL10 producing plasma B-cells in the germinal center, however if they are regarded as Bregs is still unclear (Catalan et al., 2021). In general, Bregs cannot be reliably stained with IHC/IF due the lack of a definitive set of phenotypic markers and therefore was not included in the tissue analyses. This is a problem for the field that will need to be solved in the future. We therefore used RNA Seq data to be able to analyze Breg signatures, just like other articles have done when studying this (eg. Meylan M, et al.  Early hepatic lesions display immature tertiary lymphoid structures and show elevated expression of immune inhibitory and immunosuppressive molecules. Clin Cancer Res. 2020, 26, 438–9). What we could see however is that the B cells alone are not prognostic (this was mentioned on row 361 pg 11 in original submission: Infiltrating B cells as only variable (CD20), did however not impact on survival (BCSS P= 0.38; RFI P= 0.82)).

- Authors stress that CD169 TAMs are associated with mature TLS formation. Does it imply that it is possible to find non-mature TLS (or other different stages) without CD169 TAM? Is it known which event is causal, CD169 or TLS?

RNA-Seq data was performed to verify whether TLLS indeed could be seen as bona fide TLS. This was done using RNA-Seq data showing that CD169 associate with bona fide mature TLS gene signatures. Hence, non-mature TLS is less likely but cannot be excluded fully. We believe that the addition of the RNA Seq data using mature TLS signatures in relation to CD169 is a good choice of method to answer this question and sincerely wish that referee 3 will accept this as choice of method.

The data regarding which event is causal for CD169 or TLLS is shown and discussed in Supplementary Fig 3 and 4 indicating that TLLS has causal effects over CD169 in the PT.

- In the discussion, authors bring together TILs and TLS. It seems clear that there is both a spatial and temporal dependence in the cascade of events, as high TILs count is generally a favourable features for patients, while the data presented by the authors seem to indicate that CD169 TAMs and TLLS are detrimental. Do authors have TILs count for the patients in their cohort as an additional variable for their analysis? Also, does it matter how Tregs and TLLS are spatially distributed and communicate with each other? As the count is made through pure H/E, while a TLS would require a series of staining, how do the author reason about TILs and TLS?

Thank you for this comment. We have scored B cells alone, CD3 T cells and FoxP3 Tregs. In this manuscript we show that B cells alone are not prognostic, and we have previously shown that only Tregs, but not pan T cells, are prognostic in this cohort (Stenstrom et al., 2021). The confusion regarding TILs and TLLS is acknowledged and we have now changed the phrasing on row 471 from “following high TIL levels (TLLS formation)” to “following high levels of TLLS formation” and on row 484 from “TILs” to “TLLS”.

We have not performed co-stainings with FoxP3 and TLLS so we cannot say whether Treg and TLLS are in direct spatial contact. We can however say that they are in close spatial contact when present, since FoxP3+ cells associate with TLLS in the same consecutive sectioned tumor cores (800 um diameter) and since PT lacking Treg infiltration also always lacked CD169+/TLLS dual infiltration (row 400 in original manuscript).

- The relationship between CD169 in PT vs LN met and its prognostic value is really interesting. It made wonder whether this could actually be due to a phenomenon recently described by Reticker-Flynn et al (Cell, 2022). In this work, authors provide data supporting the hypothesis that tumor cell infiltration to LN is not directly required for metastasis but rather for the generation of a local immune tolerance (through chronic IFN-gamma and epigenetic rewiring), that can be later spread systemically by Tregs. Eventually, this mechanism enables the outgrowth of distant metastases.  Can authors speculate about it?

Thank you this is a very good comment that we have now added in the discussion (with reference) on row 475-480.

Text format&typos:

Noted and corrected

- line 138 should read 800 um and not mm

- line 452 should read starting at the TLS sites  and not starting the TLS sites

CATALAN, D., MANSILLA, M. A., FERRIER, A., SOTO, L., OLEINIKA, K., AGUILLON, J. C. & ARAVENA, O. 2021. Immunosuppressive Mechanisms of Regulatory B Cells. Front Immunol, 12, 611795.

CHEN, X., ZHENG, Y., LIU, S., YU, W. & LIU, Z. 2022. CD169(+) subcapsular sinus macrophage-derived microvesicles are associated with light zone follicular dendritic cells. Eur J Immunol, 52, 1581-1594.

STENSTROM, J., HEDENFALK, I. & HAGERLING, C. 2021. Regulatory T lymphocyte infiltration in metastatic breast cancer-an independent prognostic factor that changes with tumor progression. Breast Cancer Res, 23, 27.

Round 2

Reviewer 3 Report

I would like to thank the authors for the comprehensive rebuttal letter which cleared out the few comments I raised following their initial submission. Together with the comments of the other reviewers, I believe the manuscript is ready for publication and to have its impact on the scientific community.

Good luck with your future science!